# Trends in erythemal doses at the Polish Polar Station, Hornsund, Svalbard based on the homogenized measurements (1996-2016) and reconstructed data (1983-1995)

Janusz W. Krzyścin[1], Piotr Sobolewski[1]

[1]Institute of Geophysics, Polish Academy of Sciences, Warsaw, 01-452, Poland

*Correspondence to*: Janusz W. Krzyścin (jkrzys@igf.edu.pl)

**Abstract**. Erythemal daily doses measured at the Polish Polar Station, Hornsund (77°00'N, 15°33'E), for the period 1996-2001 and 2005-2016 are homogenized using yearly calibration constants derived from the comparison of observed doses for cloudless conditions with the corresponding doses calculated by radiative transfer (RT) simulations. Modeled all-sky doses are calculated by the multiplication of cloudless RT doses by the empirical cloud modification factor dependent on the daily sunshine duration. An all-sky model is built using daily erythemal doses measured in the period 2005-2006-2007. The model is verified by comparisons with the 1996-1997-1998 and 2009-2010-2011 measured data. The daily doses since 1983 (beginning of the proxy data) are reconstructed using the all-sky model with the historical data of the column ozone from satellite measurements (SBUV merged ozone data set), the snow depth (for ground albedo estimation), and the observed daily sunshine duration at the site. Trend analyses of the monthly and yearly time series comprising of the reconstructed and observed doses do not reveal statistically significant trend in the period 1983-2016. The trends based on the observed data only (1996-2001 and 2005-2016) show declining tendency (-1%/yr) in the monthly mean of daily erythemal doses in May and June, and in the yearly sum of daily erythemal doses. An analysis of sources of the yearly dose variability since 1983 provides that cloud cover changes are a basic driver of the long-term UV changes at the site.

## 1 Introduction

The importance of the solar UV radiation on human health and ecosystems is widely discussed in the literature since the ozone hole discovery in the early 1980s (e.g. WMO, 2014). The Montreal Protocol was signed by UN countries in 1987 to protect the ozone layer, which acts as a shield against the solar UV. Since 1980 especially large ozone depletion was observed every year in the late winter and spring, the so-called ozone hole, over Antarctica (e.g. WMO, 2014). However, severe ozone losses appeared ocassionally over the Arctic, e.g. in 2011 (Garcia, 2011; Bernhard et al., 2013) and in 2016 (http://www.ametsoc.net/sotc2016/Ch05_Arctic.pdf). The ozone downward trend and the increase of the surface UV in the

Arctic was observed in 1990s (Fioletov et al., 1997; Newmann et al., 1997; Gurney, 1998).

The amount of column ozone and its vertical distribution have been measured using a ground-based and satellite network. Nowadays, the ozone distribution over the whole globe is available for scientific purposes. The surface UV radiation in the UV-B range also depends on the Sun's elevation, cloud/aerosol characteristics, and the surface albedo, which are widely variable from site to site. There are a limited number of ground-based stations measuring erythemaly effective doses continuously for longer than 20 years. These include only 5 northernmost stations above 70° N: Alert (82.5° N, 62.31° E), Ny-Ålesund (78.92° N, 11.92° E), Hornsund (77.0° N, 15.33° E), Resolute (74.72° N, 94.98° W, and Barrow (71.32° N, 156.68° W). The algorithm to calculate the surface UV using the satellite data (total ozone, ground-reflectivity) over high-latitude regions sometimes failed due to the fact that the observed high reflectivity surfaces might be erroneously classified as high ground-albedo (from snow and ice cover) or cloud effect (Tanskanen et al. 2007). It is crucial to examine the UV variability over the Arctic regions, especially for high latitudinal coastal sites, because of rich and diverse ecosystems located in this area (Hessen et al, 2001). It is anticipated that anthropogenic climate effects will be the most pronounced in high latitudinal regions (Taalas et al., 2000; IPCC 2014).

Maintaining homogeneity of long-term UV time series (20+yr) taken from various instruments is a challenging task especially for remote sites. In this paper, we propose a method for the UV data homogenization applicable for any remote Arctic site like the Polish Polar Station Hornsund (Section 3). Next, we reconstruct the UV doses dating back to 1983 when the observations of proxies for the UV variability started at Hornsund (Section 4). Finally, we search for linear trends and their uncertainty using a Monte-Carlo approach (Section 5) applied to monthly and yearly doses (Section 6) based on the reconstructed and observed data (1983-2016) and the observed data only (1996-2016 with the 2002-2004 gap).

**2 UV and ancillary data**

The erythemal UV measurements at Hornsund were carried out since 1996 up to 2001 by an improved version (with temperature stabilization) of the classic Robertson-Berger UV meter. This was a prototype of the presently widely used broadband Solar Light Model 500 (denoted SL 500) radiometer produced by Solar Light Co. RB meter was designed in the early 1970s to measure erythemal solar irradiation as its spectral characteristic resembled that of the human skin (McKinlay and Diffey, 1987). The prototype was designed in the Institute of Geophysics (IG), Polish Academy of Sciences (PAS), Belsk, in the late 1980s and since then took part in the UV monitoring at the Central Geophysical Laboratory, IG PAS, Belsk, Poland. It was moved to the Hornsund observatory in 1995 and put into regular UV monitoring in 1996 (Krzyścin and Sobolewski, 2001) that lasted up to autumn 2001. Since spring 2004, a new UV broadband meter Kipp and Zonen UVS-AE-T (Fig. 1) has been installed at Hornsund and started continuous UV monitoring in April 2005. In spring 2006 and 2007, it was calibrated by the IG PAS substandard Kipp & Zonen UVS-AE-T (No. 616), which was frequently adjusted to the Belsk's Brewer spectrophotometer (Sobolewski and Krzyścin, 2006). There were logistical difficulties with the calibration of

the Hornsund meter by a higher-level standard (e.g. the Brewer spectrophotometer) as the station could by reached only by snowmobiles (in spring), helicopters or ships (in summer). Thus, we decided to apply radiative transfer (RT) model simulations for clear-sky conditions to calibrate output of the UV-radiometer during cloudless days and perform a homogenization of the past UV data.

The following ancillary data routinely measured at Hornsund is used in the model simulations: snow depth, aerosols characteristics (aerosols optical depth, single scattering albedo from the Cimel Sunphotometer observations since 2004), and the sunshine duration (by a Campbell-Stokes recorder).

## 3 Data Homogenization

Clear-sky conditions over the Hornsund observatory were identified by the examination of the 1-minute erythemal

irradiation daily pattern. The smoothness of the pattern and the steady increase (before local noon) and decrease (after local noon) of the irradiances provided a criterion for cloudless day. The Tropospheric Ultraviolet-Visible (TUV) RT model by Madronich (1993) is implemented to calculate hypothetical clear-sky daily dose for the selected cloudless days. The TUV input consists of the column ozone amount (taken from the site overpasses by the Solar Backscatter Ultraviolet (SBUV) instrument onboard the NOAA satellites, aerosols characteristics from the AERONET database. The ground albedo in UV

range is approximated by a local formula:

$$Albedo_{GROUND} = 0.1 + Depth_{SNOW}/40, \tag{1}$$

where $Depth_{SNOW}$ is the measured snow depth in cm, $Albedo_{GROUND}$ is assumed equal to 0.9 for snow depth larger than 32 cm. Equation (1) was found experimentally, to have the best agreement with the measured daily doses in the period when the UV data at Hornsund were calibrated by the IG PAS substandard (2006-2007).

For each year (2005-2016) ratios between the modeled and observed daily doses were averaged to provide the annual correction factor, which is applied to all measured daily doses. The annual correction factor (ACF) was calculated separately for selected ranges of the noon solar zenith angle (SZA); SZA≤60°, 60°<SZA≤70°, 70°<SZA≤80°, and SZA > 80°, in the period March-June when many cloudless days were found. Figure 2 shows ACF time series (2005-2016) for four SZA ranges. It is worth noting that ACF oscillates in the range 0.95-1.05 for SZA≤60°, 60°<SZA≤70°. There is a much larger

ACF variability of ~0.75-1.35 for noon SZA>80° (early March), i.e. for a period with a weak UV intensity when the solar UV is dominated by the diffusive component related to aerosol characteristics, which sometimes is not well parameterized in the RT model. All time series shown in Fig.2 are trendless. It seems that the instrument sensitivity to the UV radiation is constant since 2005, i.e. there is no need to use any correction for the instrument aging and ACF=1.

The same procedure was used for the first period (1996-2001) of the UV monitoring at Hornsund but constant aerosols of

AOD at 340 nm equal to 0.16 was assumed. During that period there were no Cimel sunphotometer observations. Thus, for

the 1996-2001 calibration, we select AOD value representing the mean AOD value found for the period 2004-2016. Moreover, only one ACF value was calculated regardless of SZA. Figure 3 shows yearly ACF values for the period 1996-2001. It is seen that almost linear instrument deterioration of ~10% per year appeared after two years (1996-1997) of its stable behavior. The prototype instrument operated without any maintenance since 1997. Thus, it seems that the instrumental drift appeared due to increasing humidity level inside the instrument. Following ACF values, which were derived from model-observation comparisons for clear-sky days, are applied to the observed (uncorrected) daily doses: 1.01 (1996), 0.99(1997), 1.10(1998), 1.20(1999), 1.26(2000), 1.35(2001).

The uncertainty of UV observations by the prototype instrument induced by unknown aerosols AOD in ACF calculations could be estimated using extreme AOD monthly values in RT simulations, i.e. 2.5 and 97.5 percentiles of AOD values taken from all Cimel measurements in a selected month for the period 2004-2016. Figure 4 shows the differences between clear-sky erythemal daily doses calculated in 1996 for extreme high (2.5 percentile) and extreme low (97.5 percentile) AOD monthly values. Actual snow cover and satellite total ozone were used in these simulations. Because the AOD variability range depends on month, we found that the uncertainty level varies between 2-7%. Further in calculations we select 7% as a characteristic of the instrument's uncertainty induced by no precise information of aerosols loading in ACF calculations for the period 1996-2001.

## 4 Data Reconstruction

Past variations of the surface erythemal radiation in periods without UV measurements could be retrieved from statistical and radiative transfer modelling using various proxies to describe attenuation of UV radiation in the atmosphere (e.g. Lindfors and Vuilleumier, 2005; Koepke et al., 2006; Lindfors et al., 2007; Rieder et al., 2008). Global solar radiation, cloud cover, solar zenith angle, and the sunshine duration were usually used as proxies to construct empirical formulas to determine cloud attenuation of UV radiation. Junk et al. (2007) applied an advanced statistical technique, artificial neural network, to find out the most effective combination of proxies for surface UV estimation. It appeared that global solar radiation, solar zenith angles, and diffusive part of global solar radiation were essential proxies for UV reconstruction giving 1-2 percent bias and ~ 3-4% root mean square (RMS) error relative to the measured daily erythemal dose. Bilbao et al (2011) found that RMS error of ~4-9%, when only global solar irradiance and SZA were used to parameterize 10-minute erythemal doses. For some sites, only the sunshine duration was possible as a cloud attenuation proxy and it yielded 5-6% bias and RMS errors of order 20% for the reconstructed daily doses (Lindfors and Vuilleumier, 2005).

Here reconstruction of the erythemal daily doses is derived using hypothetical clear-sky erythemal daily doses from a radiative transfer model simulation using following input parameters: total ozone (for satellite overpasses), albedo (retrieved from snow depth), and aerosol optical depth at 340 nm (from collocated CIMEL sunphotometer measurements). The cloud attenuations due to clouds are derived from an empirical formula based on the daily sunshine duration measured by a

Cambell-Stokes recorder. Model's regression parameters were determined using the 2005-2006-2007 daily erythemal doses (by Kipp and Zonen UV-radiometer). The model is verified using the 1996-1997-1999 data (output of SL 500 prototype) and 2009-2010-2011 (output of Kipp and Zonen UV-radiometer).

A semi-empirical model is built to reproduce the measured daily doses (erythemal Jm$^{-2}$) for the period 2006-2007-2008.

$$DOSE_{MOD}(t) = CMF(t) \text{ x } DOSE_{CLEAR-SKY}(t) , \tag{2}$$

where $DOSE_{CLEAR-SKY}(t)$ is a hypothetical clear-sky daily dose in day $t$ from the radiative transfer model simulations (TUV) with the following input:

- the daily total ozone (TO$_3$) from the station satellite overpasses (SBUV merged data set)

- the snow albedo according to Eq.1 with the snow depth from the station meteorological data

- daily observed aerosol optical depth (AOD) at 340 nm by the collocated Cimel sunphotometer or AOD equal to 0.16, i.e., equal to long-term (2004-2016) monthly means of AOD at 340 nm, for days without CIMEL measurements

$CMF$ is an empirical cloud modification factor (CMF) used to parameterize an attenuation of hypothetical clear-sky daily doses by clouds. We have no variability of the sunshine duration throughout a day. Using the daily values adds additional uncertainties to modeled values as a duration of clear-sky conditions near local noon is decisive for daily doses.

Finally, the relative sunshine duration (in percent of the polar day durations, $SUN\_DUR(t)$, was selected as single UV regressor, and the following formula was obtained by standard least-squares approach:

$$CMF(t) = 1.0324 [SUN\_DUR(t)]^{0.1951} , \tag{3}$$

The regression coefficients (1.0324, 0.1951) are found highly statistically significant at the 99% confidence level. Model (3) explains ~ 45% of CMF variance. Table 1 shows the monthly bias and RMS errors of the model (3) performance for March – September in the period 2005-2016. The model setup is almost similar to that used by Lindords et al. (2003) for UV daily doses reconstruction for Sodankylä. However, our model provides RMS error ~ 15% for estimates of the daily erythemal dose. Lindords et al. (2003) found RMS error of ~23%.

To determine how model (2) uncertainty influences trend estimates for the whole period 1983-2016, we propose a Monte-Carlo methodology to derive the trend value and its uncertainty based on a hypothetical bootstrap sample (N=10,000) of the linear trend coefficients and their errors (see section 5).

Model (2) with CMF defined by Eq. (3) performs almost perfectly (see Fig.5a). The model-observation correlation coefficients exceed 0.9 and the smoothed pattern of scattered data obtained by LOWESS (locally weighted scatterplot smoothing, Cleveland, 1979) matches the 1-1 line (perfect agreement line - diagonal of the square). Slope by an ordinary least squares least-squares fit is 0.99 ± 0.02 (1σ), i.e., it also supports a perfect correspondence between measured and modeled daily doses.

The regression coefficient of model (2) was computed using the multi-linear least-squares fit to the observed 2005-2006-

2007 daily doses. Comparisons of the modeled data to the observed ones taken in different periods will provide a kind of the model's verification and will support a correctness of the calibration constants applied to total UV data (Section 3). Figure 5b and Figure 5c show the comparisons for the period 2009-2010-2011 and 1996-1997-1998, respectively. The model-observation correlation coefficients are high (~0.95) with linear slopes close to 1. Moreover, the smoothed patterns of scattered data points match the 1-1 line (perfect agreement line) throughout the whole range of the data variability.

The model-observation agreement appears even better for the monthly averages of daily erythemal doses (Fig.5d) for three periods together: 1996-1997-1998, 2005-2006-2007, and 2009-2010-2011. Here the correlation coefficient is equal to ~0.99 and the linear regression line has the slope of 1.002+0.009 (1σ). The simple parameterization of cloud effects on the surface erythemal dose by Eq. (3) could be used for a reconstruction of the long-term UV variability at Hornsund. Moreover, almost the same model performance was found for the periods with the UV observations done by different instruments: 1996-1997-1998 (SL 500 prototype), 2005-2006-2007 (model's built period) and 2009-2010-2011 (Kipp and Zonen UVS-AE-T radiometer). It supports the data homogeneity of the UV observations by different UV-radiometer since 1996.

## 5 Monte-Carlo method for trend estimates

We propose a Monte-Carlo procedure to estimate linear trend that accounts for various uncertainties of the daily doses throughout the examined period. The monthly and yearly trend values and their significance are derived averaging linear regression coefficients and their errors taken from a standard least-squares linear regression applied to a large number (N=10,000) of hypothetical erythemal UV time series. These time series were randomly generated for the period 1983-2016. Random representatives are generated taking into account specific uncertainty of the UV data for selected periods with different data categories, i.e.,

- for the period 1983-1995 and 2001-2004, we use the reconstructed data, based on model (2), adjusted for the model (2) uncertainty, $DOSE_{MOD, \text{ Adjusted, n}}(t)$. To the modeled value we add a random component, $RAN_n (Mean(t), SD(t))$, being $n$-th value taken from normal distribution with mean value, $Mean(t)$, and standard deviations $SD(t)$, which allows to account for possible variations of the hypothetical daily dose around its original value:

$$DOSE_{MOD, \text{ Adjusted, n}} (t) = DOSE_{MOD}(t) + RAN_n (Mean(t), SD(t)), \quad n= \{1, ... , N=10000\} \quad (4)$$

where $Mean(t)$ and $SD(t)$ are monthly mean value and pertaining standard deviation calculated from differences between the measured daily doses, $DOSE_{OBS}(t)$, in the period 2005-2016, and modeled doses, $DOSE_{MOD}(t)$, for the calendar month corresponding to $t$ value (see Table 1).

- for the period 1996-2001 and 2005-2016 we use the Monte-Carlo set of the potential representatives of the observed time series that is also adjusted for the observation uncertainty, $DOSE_{OBS \text{ Adjusted, n}} (t)$. In the former period, the data uncertainty was larger than that in the latter period because of using a non-commercial instrument and less precise

ACF calculation to account for the instrument deterioration.

$$\text{DOSE}_{\text{OBS Adjusted, n}}(t) = \text{DOSE}_{\text{OBS}}(t) + RAN_n(0, SD_k)), \qquad k=\{1, 2\}, \qquad n = \{1, \ldots, N=10000\} \qquad (5)$$

where $SD_1$ and $SD_2$ are the 1-sigma uncertainties of the daily erythemal UV measurements in the period 1996-2001 and 2005-2016, respectively. We assume that our SL prototype had an uncertainty level similar to the commercial SL UV-radiometer. Hülsen and Grobner (2007) found 11.2% and 7.2% uncertainty (at 2 sigma level) for typical SL and KZ instruments, respectively. Finally, the uncertainty of UV daily doses by the SL prototype is calculated as 13.2% taking into account 7% uncertainty induced by an assumption of a constant AOD value in the ACF calculation for the period 1996-2001 (Section 3).

10,000 hypothetical representatives of daily erythemal doses values for each day in the March-September subperiod of the period 1983-2016 were generated. Next these daily doses were averaged to produce the monthly mean of daily doses and a standard least-square linear fit was applied to each hypothetical monthly series to obtain a linear slope and its standard error. To check a hypothesis of statistical significance of the trend value at 2 sigma confidence level, we calculate a number of cases with the absolute value of the slope larger than the twice standard error of the pertaining slope. The trend is statistically significant at 2 sigma level if at least 95% of slopes fulfil this condition. The number of Monte-Carlo time series was determined by testing the stability of the mean slope and it's 2-sigma slope error when changing number of series between 1,000 and 15,000. A larger number than about 10,000 did not introduce any further changes in the statistical characteristics of the slope sample. Thus N=10,000 samples are selected for further statistical analyses.

**6 Results**

Monthly means of daily erythemal doses show significant intra-year variability (Tab.1) with the late spring early summer maximum. Therefore, to compare trend values in selected months, the trend analyses are applied to departures of the monthly mean of daily doses from pertaining the long-term monthly mean (2005-2016) in percent of the long-term monthly mean. The yearly sum of erythemal doses are calculated as a sum of the daily doses between 1st March and 30th September as earlier, and later doses are small or zero because of high solar zenith angles and polar night (between 29th October and 11th February). Similarly, to the monthly means of daily doses, the yearly sum is converted to the normalized departures relative to the mean yearly sum in the period 2005-2016.

Figure 6 illustrates the time series of the monthly (March-September) and yearly normalized departures from the long-term (2005-2016) monthly means and a yearly sum of daily doses. The regression lines show the long-term tendency in the 1983-2016 period and in the shorter period (1996-2001 and 2005-2016) when only the results of measurements were taken into account.

There are large year-to-year fluctuations in the monthly fractional deviations in the range between -40% and 40%. The linear fit to the normalized departures in the 1983-2016 period reveals a slightly increasing tendency in the yearly sums and in all

examined monthly means (excluding May with trendless behavior). The observed data for the 1996-2001 and 2005-2016 periods show a declining tendency in April, May, June, and throughout the whole year but an increasing tendency in March, August, and September. Statistical significance of the trend results is shown in Tab.2.

Table 2 presents statistical characteristics of the Monte-Carlo trend estimates. The mean linear trend coefficients and their mean standard errors together with pertaining range of estimates (between minimum and maximum of the slope and standard deviation) were derived by examining all slopes and their errors obtained by an ordinary least-square fit to each of the Monte-Carlo representative of the original time series. The trends are calculated for the period 1983-2016 (both observed and reconstructed data), for the period 1996-2016 (with observed data for the 1996-2001 and 2005-2016 periods, and the reconstructed one for the period 2002-2004), and for the period with the UV measurements only (with the gap for the period 2002-2004). The statistically significant decline at $2\sigma$ level of about -1%/yr is revealed in May, June, and in the yearly sum for the observed 1996-2016 data (with the 2002-2004 gap). The trend analyses applied to the combined observed (1996-2001 & 2005-2016) and reconstructed data (2002-2004) show statistically significant decline only in May of ~ -1% /yr.

To find sources of the long-term UV variability at Hornsund, we analyze also time series of yearly (March-September) sum of hypothetical clear-sky daily doses by RT simulations (using total ozone, aerosols, and snow albedo as model's input), and the pertaining yearly cloud modification factor (i.e., actual yearly sum divided by the corresponding clear-sky value). Figure 7a shows the yearly sum of daily doses values, for clear-sky and all-sky conditions (in $kJm^{-2}$), and CMF (in dimensionless unit). Figure 7b illustrates long-term variability of the normalized departures (% of the long term mean values for the period 2005-2015). About 10% increase of the yearly sum of all-sky daily erythemal doses and a slight decline in the clear-sky yearly sums due to the combined total ozone/albedo changes could be found in the 1980s and 1990s. Thus, it could be estimated that increasing cloud transparency and/or declining cloud cover should force ~10% an increase of yearly sums. The clear-sky UV forcing at Hornsund appears weak in the 21$^{st}$ century and UV follows mostly changes in cloud characteristics.

**7 Discussion and Conclusions**

A procedure for the examination of the UV data homogeneity is proposed based on RT simulations for clear-sky conditions. It allows introducing the yearly calibration coefficient showing the instrument sensitivity loss (1996-2001) and stable behavior in the period of measurements by the Kipp and Zonen UVS-AE-T instrument (2005-2016). For all-sky conditions, the regression model is built using 3-year data (2005-2006-2007) and comparisons of the modeled data with earlier (1996-1997-1998) and later (2009-2010-2011) data shows the same model performance as for the model building period that supports the data homogeneity and its usefulness for the long-term trend analysis. The regression model allows the UV dose reconstruction since 1983, i.e. in the period when the daily total ozone (from the satellite observations) and the sunshine duration data, which represented a proxy for the cloud effects on surface UV, were both available. The reconstruction model is also used to fill the data gaps in the UV observing period (since 1996).

Previous studies showed that the sunshine duration was a worse proxy for a parameterization of the cloud attenuation when compared to the global solar radiation. Using global solar irradiance is more appropriate to parameterize cloud effects on UV (Koepke et al., 2006). However, this variable was measured at Hornsund in some disjointed periods since 1983 but the pyranometer data were not calibrated by a higher-ranking instrument. Sunshine duration measurements by a Campbell-Stokes instrument seem to be less influenced by deterioration of the instrument's sensitivity, and its calibration is very simple as during cloudless conditions the sunburn track on a recorded cart should appear throughout the whole day.

Analyses of the yearly sums of daily erythemal doses at Hornsund reveal a non-statistically significant trend in the period 1983-2016. Two phases of the long-term behavior of total yearly doses could be identified, i.e. a positive tendency in total yearly doses in the period 1983-2000 and afterward a leveling off. The linear trend calculation by a standard least-squares fit applied to the measured (1996-2016 with the 2002-2004 gap) data shows statistically significant declining tendency in monthly means of daily doses (May and June), and in the yearly sum of the erythemal doses. However, such declining tendency is forced by two-three years of high positive fractional deviations of the erythemal doses around 2000. Longer time series (since 1983) do not show any sign of the declining tendency starting around 1996. Bernhard et al. (2011) analysis of the monthly trends at Barrow (Alaska) for the period 1990-2010 revealed statistically significant trend only in October (decline of about -1% per year) when the UV intensity was rather weak without the erythemal risk.

The stratospheric ozone changes appear as a less important driver of the UV long-term variability in the whole analyzed period. Figure 8 shows the long-term (1979-2016) pattern of the total ozone mean (using SBUV merged data) for the period May-August at Hornsund, Barrow, and Resolute, i.e. in the part of the year with naturally high UV radiation (~ 80% of total the yearly sum). The ozone forcing on the surface UV at these sites appears weak (within the ± 1% range) since 1983 (i.e. at the beginning of the reconstructed data). Cloud effects are the basic source of the UV variability at Hornsund. The albedo variations are also important as ~ 5% decline in the clear-sky modelled values in the 1980s and 1990s (Fig.7b), i.e. during the ozone declining period, seems to be forced by declining snow albedo.

It seems that the excessive UV radiation will be unlikely over the Arctic during the 21[st] century as prolonged decrease of ozone will not be possible due to the declining tendency in the concentration of the ozone-depleting chemicals in the stratosphere, anticipated intensification of the Brewer-Dobson circulation loading higher amount of ozone into the Arctic stratosphere (WMO, 2014). The downward UV tendency in the Arctic will also be induced by the increase in the cloudiness, and the lowering of the ground albedo due to the snow and sea-ice melting (e.g. Bais et al., 2015).

A continuation of UV measurements at Hornsund seems to be necessary as it is located in a region vulnerable to climate changes with the local climate strongly dependent on the heat arriving with the Gulf Stream. A projection of the weakening of the Atlantic Meridional Overturning Circulation (Boulton et al., 2014) will lead to the surface cooling at the location. It can not be excluded that high reflectivity areas (sea-ice and snow) will extend over west Svalbard and the present climatic contrast between west (warm) and east (cold) part of Svalbard will disappear. Any projection for erythemal irradiance by the end of the 21[st] century is the most uncertain for this part of the Arctic.

*Data Availability*. The total ozone overpass data was acquired from the data archive of SBUV merged ozone at ftp://toms.gsfc.nasa.gov/pub/sbuv/MERGED/. The sunshine duration and snow height at Hornsund were available at https://github.com/AtmosIGFPAN/Hornsund_Data. Aerosols optical properties were taken from AERONET database at https://aeronet.gsfc.nasa.gov/new_web/aerosols.html. Finally, the reconstructed and observed erythemal doses at Hornsund for the period 1983-2016 could be found at https://github.com/AtmosIGFPAN/Hornsund_Data.

*Acknowledgments*. Funding for this study was provided by the Ministry of Science and Higher Education, Republic of Poland, statutory activity No 3841/E-41/S/2017. We are grateful to numerous dedicated individuals who had collected meteorological data over many decades.

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

**Table 1**. **Monthly mean of the daily erythemal doses (in Joules), participation of the monthly sum of daily doses in the yearly (February-October) sum of the daily doses (in %), monthly mean difference between the observed and modelled doses (Bias in Joules) and corresponding root mean square error (RMSE in Joules), monthly mean difference between the observed and modelled doses as a percent of the observed doses (Bias in %) and corresponding root mean square error (RMSE in %). The results are from modelled and observed daily doses at Hornsund for the period 2005-2016.**

| Month | Mean [J] | % yearly dose | (Obs-Model) [ J ] | | (Obs-Mod)/Obs [ % ] | |
|---|---|---|---|---|---|---|
| | | | Bias | RMSE | Bias | RMSE |
| March | 178 | 3 | 17 | 22 | 9 | 14 |
| April | 744 | 10 | 39 | 84 | 6 | 14 |
| May | 1516 | 21 | 68 | 214 | 4 | 14 |
| June | 1983 | 27 | -6 | 237 | -0 | 14 |
| July | 1627 | 23 | -89 | 211 | -5 | 15 |
| August | 876 | 12 | -44 | 116 | -5 | 14 |
| September | 266 | 4 | -3 | 46 | -1 | 15 |

**Table 2. The monthly and yearly mean slope of the linear fit and the corresponding standard deviation (in % per year) from 10,000 sample of the hypothetical time series for the period 1983-2016 (modelled & observed data), and 1996-2016 (observed & modeled data), and for the period 1996-2001 and 2005-2016 (observed data only). The minimum and maximum values of the slope and its standard selected from 10,000 simulations are in the parentheses. Numbers in bold font represent statistically significant estimates at 2σ level.**

| Month | Slope (%/yr) | SD Error (%/yr) |
|---|---|---|
| *1983-2016 (modeled & observed)* | | |
| March | 0.46 ( 0.31,  0.32) | 0.29 ( 0.27,  0.32) |
| April | 0.23 ( 0.12,  0.37) | 0.28 ( 0.26,  0.30) |
| May | -0.04 (-0.19,  0.09) | 0.23 ( 0.20,  0.27) |
| June | 0.05 (-0.08,  0.17) | 0.27 ( 0.24,  0.29) |
| July | 0.16 ( 0.05,  0.29) | 0.29 ( 0.24,  0.31) |
| August | 0.22 ( 0.10,  0.35) | 0.23 ( 0.21,  0.25) |
| September | 0.11 (-0.03,  0.27) | 0.19 ( 0.16,  0.22) |
| Year (III-IX) | 0.11 ( 0.05,  0.17) | 0.18 ( 0.17,  0.19) |
| *1996-2016 (modeled & observed)* | | |
| March | 0.20 ( 0.00,  0.37) | 0.71 ( 0.64,  0.77) |
| April | -0.46 (-0.63, -0.29) | 0.58 ( 0.52,  0.64) |
| May | **-0.94** (-1.10, -0.75) | 0.42 ( 0.37,  0.48) |
| June | -0.98 (-1.17, -0.81) | 0.56 ( 0.50,  0.61) |
| July | 0.03 (-0.15,  0.22) | 0.45 ( 0.41,  0.49) |
| August | 0.59 ( 0.45,  0.74) | 0.44 ( 0.40,  0.49) |
| September | 0.05 (-0.17,  0.22) | 0.40 ( 0.33,  0.46) |
| Year (III-IX) | 0.40 (-0.48, -0.33) | 0.35 ( 0.33,  0.38) |
| *1996-2001 : 2005-2016 (observed)* | | |
| March | -0.17 (-0.37, -0.00) | 0.64 ( 0.57,  0.75) |
| April | 0.82 (-0.98, -0.64) | 0.47 ( 0.43,  0.51) |
| May | **-1.10** (-1.28, -0.92) | 0.38 ( 0.34,  0.43) |
| June | **-1.22** (-1.41, -1.02) | 0.50 ( 0.45,  0.55) |
| July | -0.78 (-1.11, -0.38) | 0.56 ( 0.47,  0.71) |
| August | 0.48 ( 0.33,  0.61) | 0.44 ( 0.39,  0.47) |
| September | 0.11 (-0.07,  0.28) | 0.42 ( 0.37,  0.46) |
| Year (III-IX) | **-0.75** (-0.85, -0.65) | 0.32 ( 0.30,  0.36) |

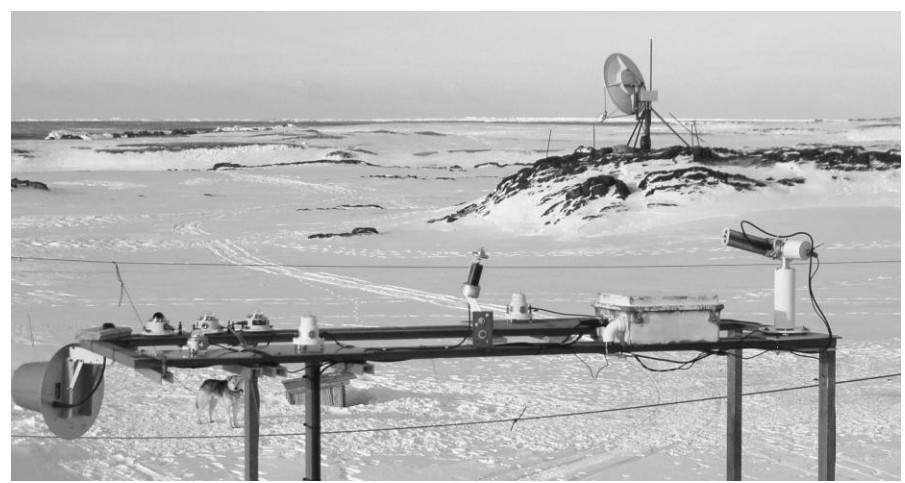

**Figure 1. The observing platform at the Polish Polar Station Hornsund (77º00'N, 15º33'E). (Photo by P. Sobolewski).**

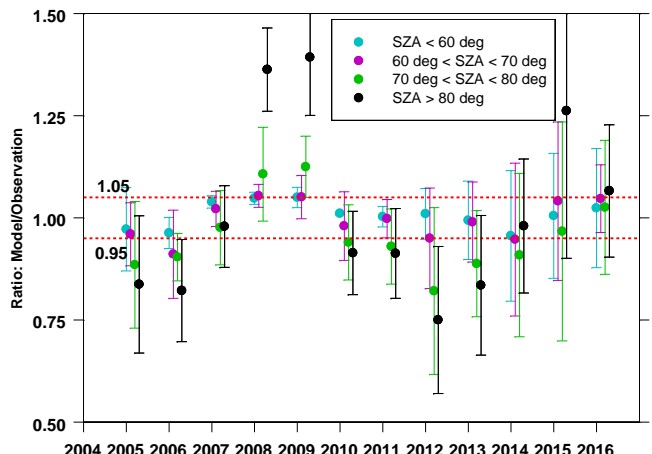

**Figure 2. The calibration constants for the Kipp and Zonen UVS-AE-T instrument in the period 2005-2016 derived from a comparison of the modeled clear sky doses with the observed ones in cloudless conditions for four solar zenith angle (SZA) ranges**

25

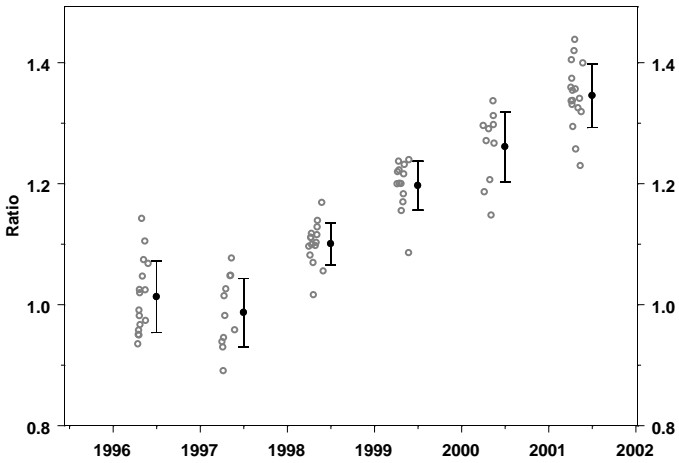

Figure 3. The calibration constants for the prototype of Solar Light instrument in the period 1996-2001 derived from a comparison of the modeled clear-sky doses with fixed aerosols optical depth (0.16 at 340 nm) with the observed ones in cloudless conditions in the March-June period. Full circles and bars represent the mean value and ± 1 standard deviation range in the selected year.

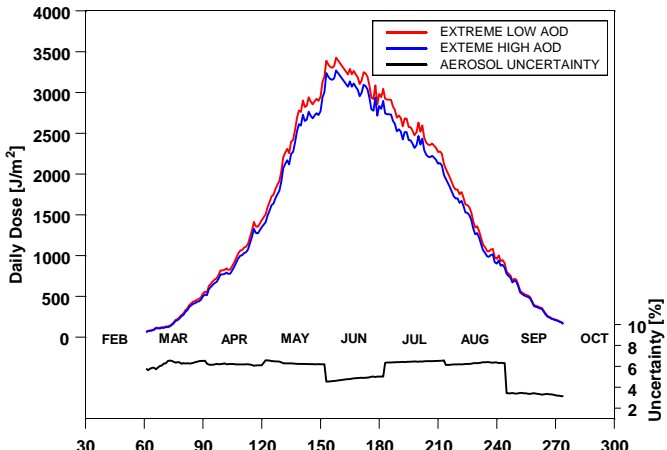

**Figure 4**. **Radiative model simulations of daily erythemal doses for clear-sky conditions in 1996 using observed total ozone, surface albedo, and extreme high and low aerosols monthly optical thickness derived from all CIMEL sunphotometer measurements at Hornsund in the period 2004-2016. Uncertainty is calculated as difference between the extreme daily doses expressed in percent of the mean daily dose.**

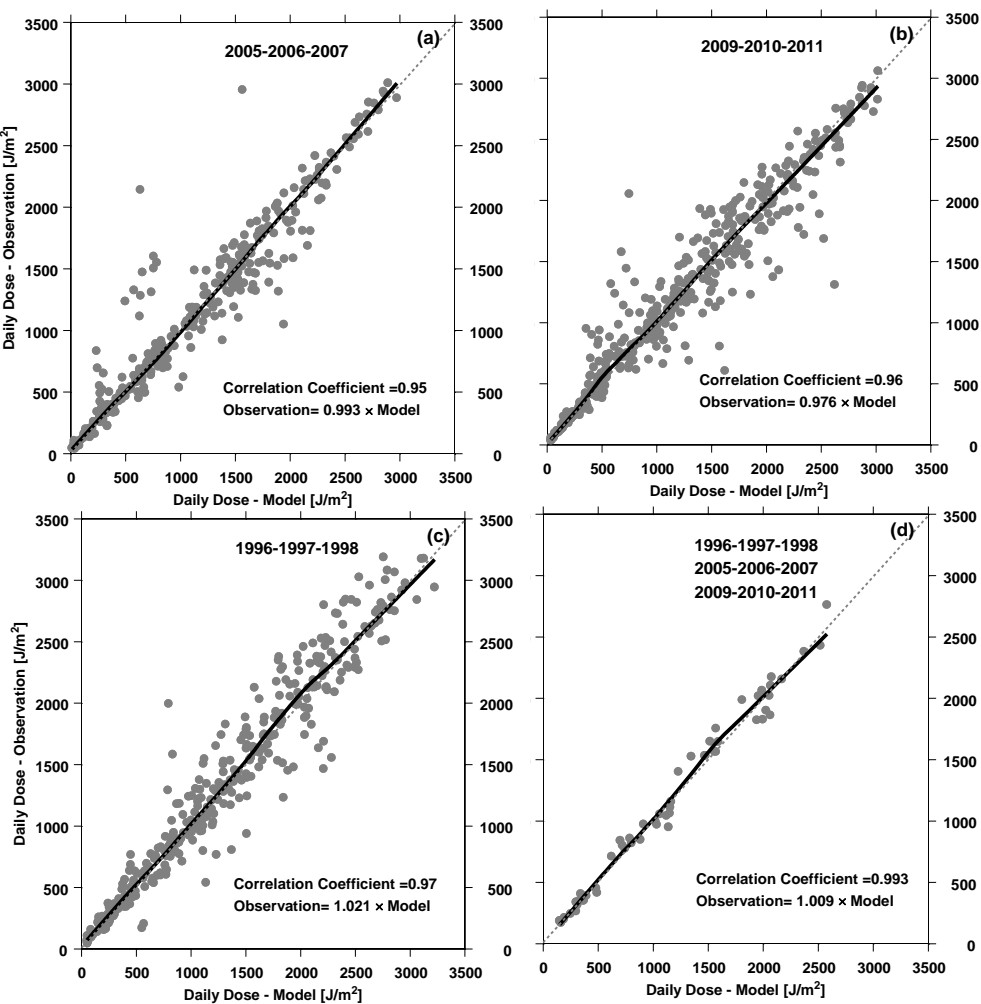

**Figure 5**. **The observed versus modeled erythemal doses: (a) daily doses for the period 2005-2006-2007, (b) daily doses for the period 2009-2010-2011, (c) daily doses for the period 1996-1997-1998, (d) monthly mean daily doses for the periods 1996-1997-1998, 2005-2006-2007, and 2009-2010-2011.**

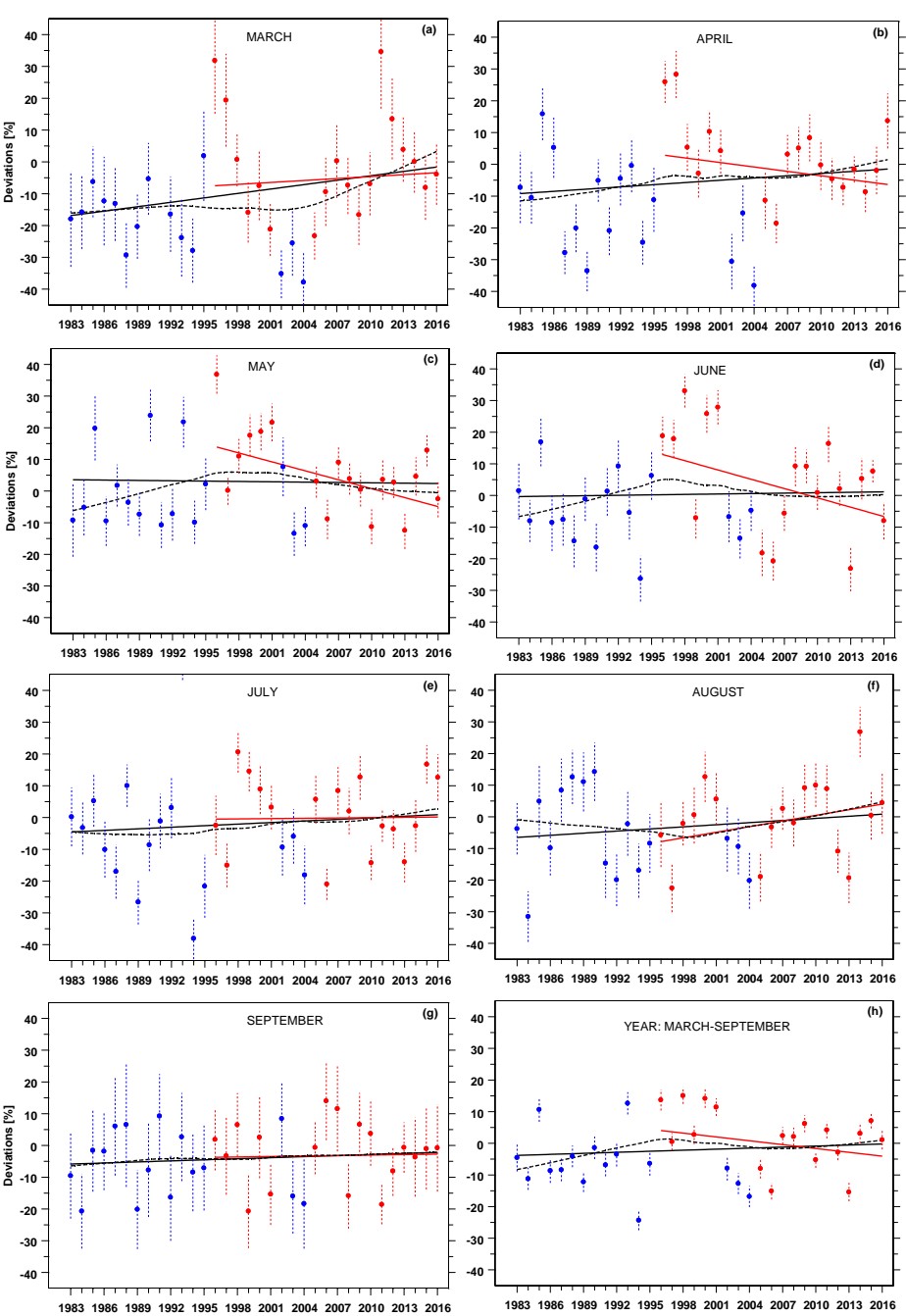

**Figure 6. Time series of normalized deviations of monthly means of daily doses and normalized yearly sum of daily doses: observed values (red circles), reconstructed values (blue circles). Vertical line shows +/- 1 standard deviation. Straight line represents linear regression line. Dashed curve represents smoothed values by LOWES smoother (Cleveland, 1979).**

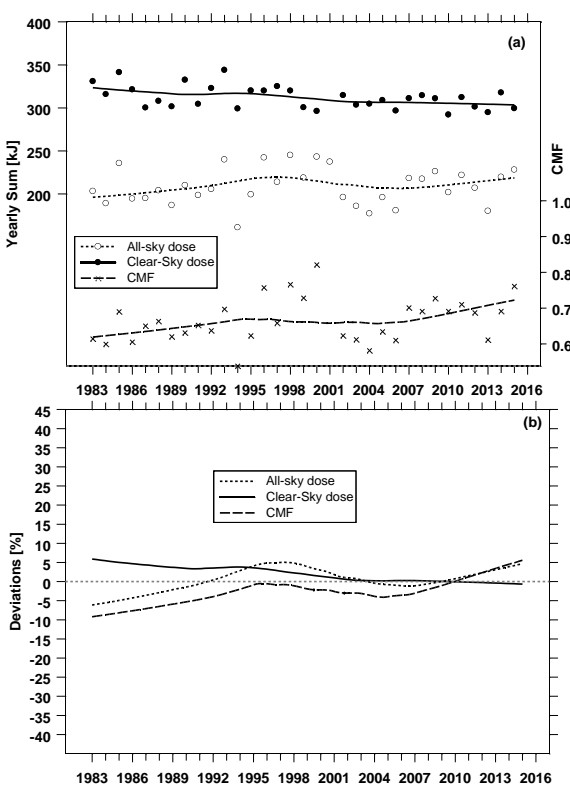

**Figure 7. (a) yearly (March-September) daily sum of erythemal doses in the period 1983-2016: modeled clear-sky values (full circles), observed and modeled all-sky values (open circles), and cloud modification factor (CMF, crosses), (b) normalized smoothed deviations (by LOWES smoother, Cleveland, 1979) of the yearly values shown in Fig.7a.**

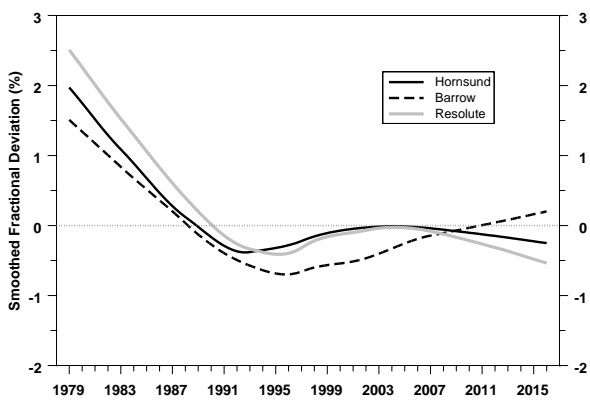

**Figure 8. Smoothed time series (by LOWES smoother, Cleveland, 1979) of annual fractional deviations of mean total ozone in period May-August for the period 1979-2016 at Hornsund (Svalbard), Barrow (Alaska, United States), and Resolute (Cornwallis Island, Canada).**

