# Peer review of "Trends in erythemal doses at the Polish Polar Station, Hornsund, Svalbard based on the homogenized measurements (1996-2016) and reconstructed data (1983-1995)"

_Atmospheric Chemistry and Physics, 2017_

## Referee Comment (RC1) · Anonymous Referee #1 · 17 Aug 2017

General comments:

The paper presents a study where measured and reconstructed data of erythemal daily doses in the Arctic station of Hornsund in Svalbard are used to estimate and attribute trends of UV radiation. The reconstructed data were obtained by simple model regressions using data of total ozone and sunshine duration as well as radiative transfer calculations for clear skies. The paper contains useful data and analyses which contribute e to understanding and quantifying the UV variability over the Arctic where measurements are sparse. The quality of the data is not very good and the authors have tried to correct using model calculations. This is not a widely accepted and recommended method and involves great uncertainties, but it can be accepted due to the uniqueness of the location and the scarcity of similar data over the area. In general I miss some estimation and discussion of the uncertainties of the derived results, particularly in estimated derived from the "homogenized" and reconstructed data.

In some parts the discussion is not very clear and should be improved and clarified with some more details.

I miss in the introduction (and possibly on results) section some discussion on reconstruction methods and their uncertainties citing relevant studies appeared in the last 10 years, as for example:

Junk, J. et al.: Reconstruction of daily solar UV irradiation from 1893 to 2002 in Potsdam, Germany, Int J Biometeorol, DOI 10.1007/s00484-00007-00089-00484, 2007.

Lindfors, A. et al.: A method for reconstruction of past UV radiation based on radiative transfer modeling: Applied to four stations in northern Europe, J. Geophys. Res., 112, D23201, doi:23210.21029/22007JD008454, 2007.

Feister, U. et al.: Long-term solar UV radiation reconstructed by ANN modelling with emphasis on spatial characteristics of input data, Atmos. Chem. Phys., 8, 3107, 2008.

Rieder, H. E. et al: Reconstruction of erythemal UV-doses for two stations in Austria: a comparison between alpine and urban regions, Atmos. Chem. Phys., 8, 6309, 2008.

Bilbao, J. et al.: Long-term solar erythemal UV irradiance data reconstruction in Spain using a semiempirical method, J. Geophys. Res., 116, D22211, 10.1029/2011jd015836, 2011.

Lindfors, A., and Vuilleumier, L.: Erythemal UV at Davos (Switzerland), 1926-2003,

estimated using total ozone, sunshine duration, and snow depth, J. Geophys. Res-Atmos., 110, D02104, doi:02110.01029/02004JD005231, 2005.

I consider the title too long: A possible alternative: "Trends in erythemal doses at the Polish Polar Station, Hornsund, Svalbard, based on homogenized measurements (1996-2016) and reconstructed data (1983-1995)"

Although I have tried to mark some of the language errors in the technical comments section, I suggest that the language should be checked again and improved. This will make the paper easier to read.

I believe that the paper can be accepted for publication after sufficient revisions and clarifications of the uncertain parts that are mentioned in the specific comments below:

Specific comments:

1, 27: I would suggest using the term severe ozone loss (or depletion) instead of "ozone hole".

3, 18: Please check this sentence: "Albedoground =0.9"?

3, 24: Why there is no plot shown for the first period? The correction factors are very large and it would be good to see them in a graph as for the second period, together with the respective standard deviations. Are there any indications in the literature for such rapid deterioration of the sensitivity of RB instruments in five years?

3, 27: How large can be the effect of aerosols at that latitude? This can be estimated with the model for the extreme climatological aerosol data of the Cimel. Then it can be inferred whether aerosols are responsible for the differences, or simply the selection of clear-sky data at high SZAs during spring and autumn months.

4, 2: I find too risky to relay the calculation of trends on data which come from an instrument with such large deterioration. Moreover, for such large year-to-year differences, monthly ACFs would have been more appropriate than yearly.

4, 10: Please mention that by using daily averages for the proxies, it is assumed implicitly that any diurnal variation of erythemal irradiance due to these proxies is not taken into account. Of course this adds to the uncertainty of the estimated daily doses.

4, 17: Please specify where the default aerosol optical depth of 0.16 is coming from.

4, 25-26: If model (3) explains less than half (45%) of the CMF variance then the reconstructed daily doses by (2) should be very uncertain, despite the highly significant regression coefficients. Please elaborate on this in the text, because if the above argument is true, then the results presented later are questionable.

5, 21: I suggest drawing on these figures the linear regressions for the whole period and the observations period with different types of lines, to support the discussion of the linear trends.

6, 2: Please mention whether the negative trends in April-May are statistically significant?

6, 4: Please make clear that by "short period" you mean the period after 1996.

6, 7: It is not clear how the weights were derived and used. Do I understand correctly that the total ozone data and the sun shine duration data were weighted with weights derived from the measured monthly erythemal doses? Moreover, is the yearly dose derived from the 12 months of only from March to September? Please make this section clearer.

6, 10: Isn't there a circular effect? The data used for the reconstruction were based on TUV calculations which used the measured total ozone, and were adjusted by the CMF which was derived by sunshine duration to account for cloud effects. Therefore, FD_TYD includes already the measured total ozone and the measured sunshine duration, and here it is regressed again against total ozone FD_TO3 and the sunshine duration FD_SUN_DUR. Please explain and clarify the discussion if I got it wrongly.

6, 14-15: This statement (full ozone recovery in 2016) is a bit strong, as the data

presented are weighted averages of ozone. As it is not clear (see previous comment) how the weights are derived and applied, this should be written more carefully.

7, 7: Please mention the statistical significance of the linear trends.

17, 6: Please mention the type of filter used for the smoothing.

Technical comments:

4, 5-6: replace "a cloud cover" by "the cloud cover" and "a sunshine" by "the sunshine"

5, 3: Replace "models (2)" with "model (2)" (singular)

5, 25-27: replace "the trendless" by "a trendless", "the decrease/increase" by "a decrease/increase" and "the turning point" with "a turning point"

5, 30: Replace "The" with "A"

6, 1-2: Delete "The" (Negative trends ∼1% . . .)

6, 13: Replace "provides" with "suggests"

6, 22: Replace "by the instrument sensitivity lost" with "by deterioration of the instrument's sensitivity"

11, 1: Specify what the bold numbers denote.

14, 10 "monthly doses for the period"; use plural (periods)
* * *

---

## Referee Comment (RC2) · Anonymous Referee #2 · 25 Aug 2017

The paper presents and analyses one of the longest time series of erythemally weighted UV data for the Arctic, based on UV-measurements and reconstructed data. The authors have utilized methods for homogenizing UV-measurements, and applied multivariate regression tools to develop models for surface albedo, cloud optical depth, as well as having validated the models against measurements. The work is scientifi-

cally relevant and interesting, as it covers a period before and after the implementation of international climate initiatives, for a pristine location.

General comments: The use of sunshine duration and snow depth data for reconstructing long term time series of UV have also been used by other authors, e.g. Lindfors et al. I miss a more extensive reference list. The calibration constants shown in Figure 2, shows that the instrument used in the period after 2004 has been quite stable, considering the harsh environment. The older instrument shows very high annual drift (factor 2.5 over 5 years). I miss some uncertainty estimates for the measurements series and reconstructed time series. A reference to Fig5.b is missing in the paper. Figure 5b: If I have understood correctly, the curves in Figure 5b are showing the yearly deviation (residuals) from the mean for the whole period, which means that the two curves labelled "Observed" and "Model" are showing the relative differences in yearly UV doses from their respective means. It would be interesting discussing the differences between real UV observations and modelled UV data. The curve labelled "Observed" is in fact a combination of reconstructed, and measured with gaps complemented with reconstructed UV data, for the whole period 1983-2016. A distinction would be appropriate, e.g. by changing the legend "Observed" to "Combined Observed and Modelled", or adding a curve with UV observations alone. Otherwise, one may think the two curves were completely independent on each other. It would also be informative for a reader to see the fractions of the monthly or yearly doses that actually were based on measurements (and not substituted with modelled data). Furthermore (figure 5b), it appears strange that the two curves labelled "Observed" and "Model" are distinctively different for periods where UV observation are missing (1983-1995 and 2002-2004), considering that both are modelled, taking the same input parameters. An explanation would be helpful for the reader.

Minor comments: Page 1, line 27: ÂńThe ozone hole over the Arctic was observed only once in 2011Âż. Even though the ozone layer was record low in the Arctic in 2011, large negative anomalies in total ozone has happened before and after 2011, e.g. in

winter 2016/17, see e.g. "State of the Climate 2016", section J: page S151-S154. http://www.ametsoc.net/sotc2016/Ch05_Arctic.pdf Please, consider a reformulation.

Page 2, line 29-31: "During the two years of its operation...". A reader may first believe the instrument was operating only for two years. The meaning is likely rather "During 2006 and 2007 the instrument was calibrated...".

Page 3 line 4: "Biometer" is normally associated with another brand of erythemal UV radiometers; the Solar Light Co. UV-Biometer. Please, consider using the wording UV-radiometer instead, for all instances of "biometer".

Page 3 line 18, There should likely be a comma instead of a dot (.) after ">32 cm".

Section 5 Results and section 6 Discussion and Conclusion: Please, consider restructuring, or moving overlapping information. Example: page 6 lines 1-5 is restated on page 7 lines 4-9. Information on page 6-13 could be moved to the materials sections. Information on page 6 lines 17-24 could be moved to the Discussions section.

Page 7 line 14: Belsk should probably be Hornsund. Page 14, legend to Figure 3d: "Monthly doses" are probably monthly mean daily doses.

---

## Referee Comment (RC3) · Anonymous Referee #3 · 5 Sep 2017

The paper discusses the calibration of a long-term surface UV record and erythemal dose (measured or modeled) at a high northern latitude site, Hornsbud, in Svalbard. The surface record is obtained from different ground-based instruments that are impacted by different levels of instrument degradation. There are also gaps in the observational record. A model of erythemal dose has been used, in comparison against

measurements, to derive annual correction factors to bring the components of the instrumental record on to the same scale for long-term studies. Then, from the homogeneous long-term record, trend analysis of surface UV radiation is performed and linear regression analysis is utilized to attribute changes in long-term trends to physical features, such as clouds.

In general, I feel the paper could be strengthened by discussion of the uncertainties in the results. This might require additional calculations that address sensitivities in the derived results to assumptions in the corrections. Uncertainty bars would be very beneficial for the trend analysis discussion. The discussion of the approach to homogenize the observed data for $\sim$ 20 years from the high-latitude station is of benefit.

General comments on instrument correction/calibration:

Attempts to correct a long-term instrumental record for instrument artifacts is a valuable contribution given the sparsity of ground-based UV radiation and erythemal dose measurements, in particular in the Arctic, where high-latitude retrievals of these variables from satellite observations is challenging due to difficulty in separating bright surface from cloud effects. An annual correction factor (ACF), to correct periods of the instrumental record, such that ratios of modeled to measured erythemal daily dose are $\sim$ within +/- 5% (at solar zenith angles $\sim$ 60 to 70 degrees for the time range 2004-2016.

I do not find in the discussion of the Annual Correction Factor, for the 5 year time period from 1996 to 2001, why the ACF value is so large and reaches a factor of 2.5 over five years. Is that a typical degree of instrument degradation for the Robertson-Berger UV meter? I also miss how sensitive the ACF value is to assumed AOD value of 0.16 and to assumption of no dependency on solar zenith angle. Additionally, please clarify what is the time period over which an assumed AOD of 0.16 is assumed: is it 1996-2001 (p.3, l.30) or 2004-2014 (p.4, l.16). I think more discussion of this result and the implication of the degree to which the trend analysis of the long-term record will be subsequently affected by derived ACF factor is required because there is an obvious

"knee-bone" around 2006 in the erythemal dosage time series in Figures 4 and 5b. A sensitivity analysis to incremental changes in assumed AOD could be performed at the very least to provide some uncertainty around the ACF value.

I also do not find if (and how) uncertainty in the ACF is propagated into the coefficients derived from the linear regression analysis.

General comments on proxy model approach for daily erythemal dose: A proxy model is derived to extend modeled (using TUVS model) surface UV for clear sky between 20065 and 2008 to all sky conditions back to 1983. The proxy model is compared against measurement record in 1996-1999 (corrected by ACF) and 2009-2011 (where ACF = unity). The relationship between US and erythemal dose is a function of ozone column, surface albedo, aerosols and clouds. The TUVS model has the first three as inputs from satellite observations, a parameterized albedo model as a function of snow depth, and aerosol observations. An empirical factor, a function of sunshine duration, is applied to account for clouds. Clouds, due to their temporal and spatial variability, and changing optical properties as a function of low (predominantly water) and high (predominantly ice) altitude will be difficult to proxy model well.

I cannot understand how the sunshine duration, as a proxy of clouds, is found to be highly statistically significant, when this approach is found to explain only 45% of the cloud modification? What was the criteria that was used to select sun duration as the best regressor for clouds? A correlation coefficient of greater than 0.9 is reported when regressing modeled and measured erythemal doses (Fig 3). I do not find the sigma (uncertainty in the regression best fit line) reported. What uncertainty is assumed/applied for the observed daily erythemal dose in the regression? While standard linear regression does not allow for uncertainties in the regressor, a somewhat related approach called Orthogonal distance regression (ODR) does. I find that clarification and additional discussion about the uncertainty in the proxy model regression to derive the cloud modification factor is required. An assessment of the propagation of this uncertainty into trend analysis would be helpful. Perhaps an ODR approach could contribute

to an improved understanding of the sensitivity in the derived scaling coefficients to uncertainties in the modeled erythemal dose.

If I understand correctly, a second proxy model of total yearly dose of erythemal radiation is derived from a linear regression of the fractional deviation in yearly dose, where the model contribution in this fractional deviation comes from another multiple linear regression proxy model incorporating sunshine duration. I am not aware of "nested" multiple linear regression proxy models in general. Is this a commonly applied approach and are their references that can be cited as examples? I would feel that the uncertainties from the first proxy model would propagage into uncertainties in the second proxy model (and likely not in a linear fashion due to the nonlinear behavior between clouds, ozone, surface albedo and radiation). Some discussion and acknowledgement of the potential pitfalls of this approach would be helpful in the paper.

General comments on trend analysis: The proxy model (Eqtn 2) will be sensitive to clouds, as discussed in the paper. An underlying change in cloud fractions, cloud type (altitude, thermodynamic phase) over long time periods will manifest in the observed surface UV but will not be captured by the proxy model. Therefore, I find that ascribing behavior in long-term trends using the described approach somewhat dangerous, in particular given the large amount of uncertainties inherent in the approach for empirical cloud modification. The analysis that the conclusions are drawn from should really contain uncertainty bars to guide the interpretation of the concluding statements regarding trends in ozone and cloudiness.

---

## Author Comment (AC1) · 11 Oct 2017

Anonymous Referee #1 General comments: In general, I miss some estimation and discussion of the uncertainties of the derived results, particularly in estimated derived from the "homogenized" and reconstructed data. In the revised manuscript, the uncer-

tainty of the daily erythemal doses used in trend calculations have been calculated for the data categories: reconstructed (1983-1995, 2002-2004), measured by a prototype of SL biometer (1996-2001), and KZ biometer (2005-2016). See p.5, l.18-21; p.6-7, l.19-l.7 and Table 1.

In some parts the discussion is not very clear and should be improved and clarified with some more details. More detailed description of the trend methodology and data preparation are included in the revised manuscript I miss in the introduction (and possibly on results) section some discussion on reconstruction methods and their uncertainties citing relevant studies appeared in the last 10 years, as for example: (list of publications) This part is added to the main text (p.4, l.16-26) using the reviewer's list of most important studies. Moreover, a performance of the proposed reconstruction model is compared to the study using similar proxies for UV attenuation in the atmosphere (p.5, l.19-21).

I consider the title too long: A possible alternative: "Trends in erythemal doses at the Polish Polar Station, Hornsund, Svalbard, based on homogenized measurements (1996-2016) and reconstructed data (1983-1995)" OK. The title has been changed according the reviewer's suggestion.

Although I have tried to mark some of the language errors in the technical comments section, I suggest that the language should be checked again and improved. This will make the paper easier to read. We have tried to improve the language, for example the manuscript has been read by a foreign speaker.

Specific comments: 1, 27: I would suggest using the term severe ozone loss (or depletion) instead of "ozone hole". OK. Change according the reviewer's suggestion.

3, 18: Please check this sentence: "Albedoground =0.9"? New sentence is "AlbedoGROUND is assumed equal to 0.9 for snow depth larger than 32 cm"

3, 24: Why there is no plot shown for the first period? The correction factors are very

large and it would be good to see them in a graph as for the second period, together with the respective standard deviations. Are there any indications in the literature for such rapid deterioration of the sensitivity of RB instruments in five years? New Figure 3 is added showing the instrument deterioration. In fact, the deterioration appeared much smaller ∼35% in the period 1996-2001. The previously mentioned deterioration rate (∼250%) was erroneously calculated. WMO report (Instrument to Measure Solar Ultra-violet Radiation Part 2: Broadband Instruments Measuring Erythemally Weighted Solar Irradiance", WMO, Rep. No. 164, 2008) stated that the well maintained broadband in-strument could lost its stability maximally up to 5% between yearly intercomparisons. Thus, the loss of about 10% per year after two years of stable behavior (1996-1997) seems possible in a harsh polar environment. p. 4, l.1-6.

3, 27: How large can be the effect of aerosols at that latitude? This can be estimated with the model for the extreme climatological aerosol data of the Cimel. Then it can be inferred whether aerosols are responsible for the differences, or simply the selection of clear-sky data at high SZAs during spring and autumn months. The extreme aerosols optical depth (AOD) for each month (March-September) are determined from 2.5 and 97.5 percentiles of the daily AOD values in selected month by Cimel measurements (2004-2016). These values are used in radiative model simulations to calculate the daily dose uncertainty due to unknown AOD in period prior Cimel measurements. Un-certainty (∼7%) of the annual correction factor ACF for the period 1996-2007 is found. See p.4, l.7-14, and Figure 4.

4, 2: I find too risky to relay the calculation of trends on data which come from an instru-ment with such large deterioration. Moreover, for such large year-to-year differences, monthly ACFs would have been In fact, the year-to-year deterioration appeared smaller that discussed in the previous manuscript. It is around ∼9% per year, i.e. 9/12% per month. The yearly ACF is derived using mostly from April-June data (i.e. period with many cloudless days). Thus, the July, August, and September value could be underes-timated of ∼ 0.75%, 1.5%, and 2.25%, respectively, after application of the proposed

yearly ACF. The trend calculation for each calendar month (March-September) is not affected by ACF changes within the year. Only the yearly trend could be affected. Taking into account a participation of monthly mean doses for these months in the yearly dose (i.e., 23%, 12%, 4% for July, August, and September, respectively, see new Table 1) it could be estimated that using the yearly ACF would provide less than ∼1% underestimation of the yearly dose. Thus, using yearly rate of the instrument deterioration instead of the monthly rate affects only slightly trend estimates of the yearly sums of erythemal daily doses.

4, 10: Please mention that by using daily averages for the proxies, it is assumed implicitly that any diurnal variation of erythemal irradiance due to these proxies is not taken into account. Of course, this adds to the uncertainty of the estimated daily doses. We discuss the problem in the revised paper: "We have no variability of sunshine duration throughout a day. Using the daily values adds additional uncertainties to modeled values as a duration of clear-sky conditions near local noon is decisive for daily doses." p.5, l. 12-13.

4, 17: Please specify where the default aerosol optical depth of 0.16 is coming from. In the revised paper, we explain that " "The same procedure was used for the first period (1996-2001) of the UV monitoring at Hornsund but constant aerosols of AOD at 340 nm equal to 0.16 was assumed. During that period there were no Cimel sunphotometer observations. Thus, for the 1996-2001 calibration, we select AOD value representing the mean AOD value found for the period 2004-2016" p. 3, l.28-30.

4, 25-26: If model (3) explains less than half (45%) of the CMF variance then the reconstructed daily doses by (2) should be very uncertain, despite the highly significant regression coefficients. Please elaborate on this in the text, because if the above argument is true, then the results presented later are questionable. We agree that the explained variability of the reconstructed data is low. However, the root mean square error of the reconstructed data of ∼ 15% (see Table 1) is comparable with performance of previous reconstruction models (e.g. Lindfors et al., 2003 for Sodankyla, Rieder at

al., 2008 for Sonnblick). Moreover, the proposed Monte-Carlo procedure to calculate the trend error takes into account the uncertainty of the reconstructed daily doses.

5, 21: I suggest drawing on these figures the linear regressions for the whole period and the observations period with different types of lines, to support the discussion of the linear trends. New Figure 6 is prepared and lines are drawn for the 1983-2016 and 1996-2016 periods.

6, 2: Please mention whether the negative trends in April-May are statistically significant? In the revised manuscript we have: "The statistically significant decline at $2\sigma$ level of about 1% /yr is revealed in May, June, and in the yearly sum for the observed 1996-2016 data (with the 2002-2004 gap)." p.8, l.8-11.

6, 4: Please make clear that by "short period" you mean the period after 1996. It has been defined. See response to the previous (6.2) problem.

6, 7: It is not clear how the weights were derived and used. Do I understand correctly that the total ozone data and the sun shine duration data were weighted with weights derived from the measured monthly erythemal doses? Moreover, is the yearly dose derived from the 12 months of only from March to September? Please make this section clearer. In the revised paper the ozone effects are discussed using different approach without the above mentioned weighting. The daily doses from radiative model simulations for clear-sky conditions are used to calculate the yearly sum of the daily erythemal doses. The clear-sky data are compared with the original (modeled and measured) to discuss the cloud and ozone/albedo forcing of the UV. The yearly sum of daily doses are taken from March-September data because of small intensity in UV radiation in February and October and polar night between end of October and mid February. p 7. l.22-24.

6, 10: Isn't there a circular effect? The data used for the reconstruction were based on TUV calculations which used the measured total ozone, and were adjusted by the CMF which was derived by sunshine duration to account for cloud effects. Therefore,

FD_TYD includes already the measured total ozone and the measured sunshine duration, and here it is regressed again against total ozone FD_TO3 and the sunshine duration FD_SUN_DUR. Please explain and clarify the discussion if I got it wrongly. In the revised paper, the different method is proposed to search for clouds and ozone/albedo impact on UV. The results by radiative transfer model for clear-sky conditions are examined in the period 1983-2016. Regression using FD_TYD as a linear function of FD_TO3 and FD_SUN_DUR has been rejected.

6, 14-15: This statement (full ozone recovery in 2016) is a bit strong, as the data presented are weighted averages of ozone. As it is not clear (see previous comment) how the weights are derived and applied, this should be written more carefully. Problem of the ozone recovery is not discussed in the revised manuscript. We only say that "The stratospheric ozone changes appear as less important driver of the UV long-term variability in the whole analyzed period. Figure 8 shows the long-term (1979-2016) pattern of the total ozone mean (using SBUV merged data) for the period May-August at Hornsund, Barrow, and Resolute, i.e. in the part of the year with naturally high UV radiation ($\sim$ 80% of total yearly sum). The ozone forcing on the surface UV at these sites appears weak (within the $\pm$ 1% range) since 1983 (i.e. at the beginning of the reconstructed data)." p.9, l.16-22.

7, 7: Please mention the statistical significance of the linear trends. We define significance of the trend in the revised paper: "The statistically significant decline at $2\sigma$ level of about -1% per year is revealed in May, June, and in the yearly sum for the observed 1996-2016 data (with the 2002-2004 gap). The trend analyses applied to the combined observed (1996-2001 & 2005-2016) and reconstructed data (2002-2004) show statistically significant decline only in May of $\sim$ -1%/yr." p.8, l.8-11.

17, 6: Please mention the type of filter used for the smoothing. We explain in the revised manuscript: "Figure 8. Smoothed time series (by LOWES smoother, Cleveland, 1979) of annual fractional deviations. . . . . . .." p.22

[Figure]

Technical comments: The reviewer suggestions (below) are included in the revised manuscript" 4, 5-6: replace "a cloud cover" by "the cloud cover" and "a sunshine" by "the sunshine" 5, 3: Replace "models (2)" with "model (2)" (singular) 5, 25-27: replace "the trendless" by "a trendless", "the decrease/increase" by "a decrease/increase" and "the turning point" with "a turning point" 5, 30: Replace "The" with "A" 6, 1-2: Delete "The" (Negative trends +/-1% ...)  6, 13: Replace "provides" with "suggests" 6, 22: Replace "by the instrument sensitivity lost" with "by deterioration of the instrument's sensitivity" 11, 1: Specify what the bold numbers denote. 14, 10 "monthly doses for the period"; use plural (periods)

Revised manuscript is in the attached supplement file. Please note the change of the manuscript title according to the referee #1 comment.

Please also note the supplement to this comment:
https://www.atmos-chem-phys-discuss.net/acp-2017-619/acp-2017-619-AC1-supplement.pdf

**Supplement:**

[revised manuscript text omitted]

---

## Author Comment (AC2) · 11 Oct 2017

General comments: The use of sunshine duration and snow depth data for reconstructing long term time series of UV have also been used by other authors, e.g. Lindfors et

al. I miss a more extensive reference list. The calibration constants shown in Figure 2, shows that the instrument used in the period after 2004 has been quite stable, considering the harsh environment. The older instrument shows very high annual drift (factor 2.5 over 5 years). I miss some uncertainty estimates for the measurements series and reconstructed time series. In fact, the instrument deterioration in the period 1996-2001 appeared much smaller about 35% (not 2.5 as it was previously mentioned). We add new Figure (new Fig. 3) showing the loss of instrument sensitivity in this period. We discuss some important studies on the UV reconstruction models (the beginning of section 4). Moreover, a performance of the proposed reconstruction model is compared to the previous study (Lindfors et al., 2003) using similar proxies for UV attenuation in the atmosphere. We explain: "The model setup is almost similar to that used by Lindords et al. (2003) for UV daily doses reconstruction for Sodankylä. However, our model provides RMS error ∼15% for estimates of the daily erythemal dose. Lindords et al. (2003) found RMS error of ∼23%." p.5, 19-21.

A reference to Fig5.b is missing in the paper. Figure 5 has been replaced by Fig.7., which illustrates changes if UV radiation due to combined ozone and albedo effects (simulation by radiative transfer model for clear sky conditions). We think that new Figure illustrates better impact on cloudiness on surface UV at Hornsund.

Figure 5b: If I have understood correctly, the curves in Figure 5b are showing the yearly deviation (residuals) from the mean for the whole period, which means that the two curves labelled "Observed" and "Model" are showing the relative differences in yearly UV doses from their respective means. It would be interesting discussing the differences between real UV observations and modeled UV data. In the revised manuscript, we explain. There are not such curves in new Fig.7. Previous "modeled" curves were obtained with the regression model applied to the weighted data (Eq. 5 in the previous manuscript) that was criticized by the reviewers. The differences between the observed and modeled data (based on cloud modification factor defined by Eq. 2) are shown in new Table 1. p.13

The curve labeled "Observed" is in fact a combination of reconstructed, and measured with gaps complemented with reconstructed UV data, for the whole period 1983-2016. A distinction would be appropriate, e.g. by changing the legend "Observed" to "Combined Observed and Modelled", or adding a curve with UV observations alone. We decide to delete previous Fig.5 as it combined results of two regression models (previous Eq.2 and Eq.5) and it was difficult to find out meaning of the modeled data. Moreover, the second model was not correctly defined and it was rejected.

Otherwise, one may think the two curves were completely independent on each other. It would also be informative for a reader to see the fractions of the monthly or yearly doses that actually were based on measurements (and not substituted with modelled data). The monthly and yearly doses in the 1996-2001 and 2005-2015 periods are derived from almost every day UV measurements, so the gap existed only for period March 2002 up to April 2005. In the revised paper we calculated trends for both the 1996-2016 time series with the data holes filled by the modeled data and for the time series comprising only observations.

Furthermore (figure 5b), it appears strange that the two curves labelled "Observed" and "Model" are distinctively different for periods where UV observation are missing (1983-1995 and 2002-2004), considering that both are modelled, taking the same input parameters. An explanation would be helpful for the reader. The both curves were modeled by different models; previous Eq. 2 for cloud modification factor and previous Eq.5 for yearly sum of daily doses variability. We do not follow this concept in the revised paper. We explain the long-term cloud effects on surface UV in much simpler way.

Minor comments: Page 1, line 27: "The ozone hole over the Arctic was observed only once in 2011" Even though the ozone layer was record low in the Arctic in 2011, large negative anomalies in total ozone has happened before and after 2011, e.g. in winter 2016/17, see e.g. "State of the Climate 2016", section J: page S151-S154. http://www.ametsoc.net/sotc2016/Ch05_Arctic.pdf. Please, consider a reformulation.
* * *
Interactive
comment

We add a statement according the reviewer's comment. "However, severe ozone losses appeared occasionally over the Arctic, e.g. in 2011 (Garcia, 2011; Bernhard et al., 2013) and in 2016 (http://www.ametsoc.net/sotc2016/Ch05_Arctic.pdf)." p.1, l.25-28.

Page 2, line 29-31: "During the two years of its operation ...". A reader may first believe the instrument was operating only for two years. The meaning is likely rather "During 2006 and 2007 the instrument was calibrated. We change the text according the reviewer's comment. p.2, l 28-29.

Page 3 line 4: "Biometer" is normally associated with another brand of erythemal UV radiometers; the Solar Light Co. UV-Biometer. Please, consider using the wording UV-radiometer instead, for all instances of "biometer". "Biometer" has been replaced by "UV-radiometer" in the revised manuscript.

Page 3 line 18, There should likely be a comma instead of a dot (.) after ">32 cm". OK. It has been removed.

Section 5 Results and section 6 Discussion and Conclusion: Please, consider restructuring, or moving overlapping information. Example: page 6 lines 1-5 is restated on page 7 lines 4-9. In the revised paper in section 7 (Discussion and Conclusion) we state that "The linear trend calculation by a standard least-squares fit applied to the measured (1996-2016 with the 2002-2004 gap) data shows statistically significant declining tendency in the monthly mean of daily doses (May and June), and in the yearly sum of the erythemal doses. However, such declining tendency are forced by two-three years of high positive fractional deviations of the erythemal doses around 2000." p 9, l.7-9. According reviewer's suggestion in this scection, we cut details of the trend (overlapping information) but focus on a source of such trend behavior.

Information on page 6-13 could be moved to the materials section. This part of text has been deleted as it concerns the models' results not used in the revised manuscript.

Information on page 6 lines 17-24 could be moved to the Discussions section. OK. Now this part appears in the Discussion section.

Page 7 line 14: Belsk should probably be Hornsund. We change the text according the reviewer's comment.

Page 14, legend to Figure 3d: "Monthly doses" are probably monthly mean daily doses. We change the text according the reviewer's comment.

Please also note the supplement to this comment:
https://www.atmos-chem-phys-discuss.net/acp-2017-619/acp-2017-619-AC2-supplement.pdf

―――――――――――――――――

**Supplement:**

[revised manuscript text omitted]

---

## Author Comment (AC3) · 11 Oct 2017

Response to the reviewer's comments - Referee #3

In general, I feel the paper could be strengthened by discussion of the uncertainties in the results. This might require additional calculations that address sensitivities in the

derived results to assumptions in the corrections. The basic difference between the previous and revised manuscript is using a special Monte-Carlo procedure of generation hypothetical time series of daily erythemal doses accounting for various uncertainties for different data categories: reconstructed data, measured by the SL prototype, and measured by KZ instrument (see new section 5). The trend values are derived averaging sample of linear slopes derived by a standard least-squares fit applied to each Monte-Carlo time series.

Uncertainty bars would be very beneficial for the trend analysis discussion. The discussion of the approach to homogenize the observed data for 20 years from the high-latitude station is of benefit. The uncertainty bar appear in the revised manuscript (see new Fig. 6, p.20). Moreover, the model-observation differences are shown in new Tab.1. (p.13)

General comments on instrument correction/calibration:

I do not find in the discussion of the Annual Correction Factor, for the 5 year time period from 1996 to 2001, why the ACF value is so large and reaches a factor of 2.5 over five years. Is that a typical degree of instrument degradation for the Robertson-Berger UV meter? I also miss how sensitive the ACF value is to assumed AOD value of 0.16 and to assumption of no dependency on solar zenith angle. New Figure is added showing the instrument deterioration. In fact, the deterioration appeared much smaller ∼35% in the period 1996-2001. The previously mentioned deterioration rate (∼250%) was erroneously calculated. WMO report (Instrument to Measure Solar Ultraviolet Radiation Part 2: Broadband Instruments Measuring Erythemally Weighted Solar Irradiance", WMO, Rep. No. 164, 2008) stated that the well maintained broadband instrument could lost its stability maximally up to 5% between yearly intercomparisons. Thus, the loss of about 10 per year after two years of stable behavior (1996-1997) seems possible in a harsh polar environment. The extreme aerosols optical depth (AOD) for each month (March-September) are determined from 2.5 and 97.5 percentiles of the daily AOD values in selected month by Cimel measurements (2004-2016). These values

are used in radiative model simulations to calculate the daily dose uncertainty due to unknown AOD in period prior Cimel measurements. Uncertainty (∼7%) of the annual correction factor ACF for the period 1996-2007 is found. See Figure 4. p.18.

Additionally, please clarify what is the time period over which an assumed AOD of 0.16 is assumed: is it 1996-2001 (p.3, l.30) or 2004-2014 (p.4, l.16). In the revised manuscript we explain that "... for the 1996-2001 calibration, we select AOD value representing the mean AOD value found for the period 2004-2016" p.3., l 30. We used this value also if there were no observations of AOD because of bad weather. "daily observed aerosol optical depth (AOD) at 340 nm by the collocated Cimel sunphotometer or AOD equal to 0.16, i.e., equal to long-term (2004-2016) monthly means of AOD at 340 nm, for days without CIMEL measurements " p.5, l. 9-10.

I think more discussion of this result and the implication of the degree to which the trend analysis of the long-term record will be subsequently affected by derived ACF factor is required because there is an obvious knee-bone" around 2006 in the erythemal dosage time series in Figures 4 and 5b. A sensitivity analysis to incremental changes in assumed AOD could be performed at the very least to provide some uncertainty around the ACF value. also I do not find if (and how) uncertainty in the ACF is propagated into the coefficients derived from the linear regression analysis. As we mention before, the trend are calculated using a novel trend method accounting for data uncertainties depending on data collection periods: 1983-1995 for reconstructed data, 1996-2001 for the SL prototype, 2002-2004 for the reconstructed data, and since 2005 up to the end of data for KZ data.

An empirical factor, a function of sunshine duration, is applied to account for clouds. Clouds, due to their temporal and spatial variability, and changing optical properties as a function of low (predominantly water) and high (predominantly ice) altitude will be difficult to proxy model well. Previous paper shows that the combined cloud effects on UV could be parameterized using proxies and solar duration appeared one of possible proxies (see introduction to section 4).

I cannot understand how the sunshine duration, as a proxy of clouds, is found to be highly statistically significant (our response: it means that there is strong linear dependence between the proxy and UV radiation, i.e. long sunshine duration corresponds to higher doses and zero/short duration means small UV doses), when this approach is found to explain only 45% of the cloud modification? (our response: it means that other factors are also important i.e., cloud transparency, period of the day with cloudless condition as noon conditions are decisive for the dose value)

What was the criteria that was used to select sun duration as the best regressor for clouds? We add paragraph (introduction to section 4. Data Reconstruction) providing some details of previous papers focusing on UV modeling. The sun duration was among the regressors used to explain UV variability. There were better set of the regressors (global solar radiation, diffusive component of solar radiation, etc. ) but only sunshine duration was available at Hornsund. Of course, it provide a large uncertainty of the reconstructed doses but the trends were calculated taking into the data uncertainty.

The correlation coefficient of greater than 0.9 is reported when regressing modeled and measured erythemal doses (Fig 3). I do not find the sigma (uncertainty in the regression best fit line) reported. We add following statement: "Slope by an ordinary least squares least-squares fit is $0.99 \pm 0.02$ ($1\sigma$), i.e., it also supports a perfect correspondence between measured and modeled daily doses", P.5, 28-30.

What uncertainty is assumed/applied for the observed daily erythemal dose in the regression? While standard linear regression does not allow for uncertainties in the regressor, a somewhat related approach called Orthogonal distance regression (ODR) does. I find that clarification and additional discussion about the uncertainty in the proxy model regression to derive the cloud modification factor is required. An assessment of the propagation of this uncertainty into trend analysis would be helpful. Perhaps an ODR approach could contribute to an improved understanding of the sensitivity in the derived scaling coefficients to uncertainties in the modeled erythemal dose. In

the revised manuscript, trends are estimated using Monte-Carlo approach taking into account uncertainties in the reconstructed (see new Table 1) and measured data (different values for the measurements by the prototype for the period 1996-2001, and KZ instrument for the period 2005-2016) . New section 5 (Monte-Carlo method for trend estimates) explains the methodology used.

If I understand correctly, a second proxy model of total yearly dose of erythemal radiation is derived from a linear regression of the fractional deviation in yearly dose, where the model contribution in this fractional deviation comes from another multiple linear regression proxy model incorporating sunshine duration. I am not aware of "nested" multiple linear regression proxy models in general. Is this a commonly applied approach and are their references that can be cited as examples? I would feel that the uncertainties from the first proxy model would propagate into uncertainties in the second proxy model (and likely not in a linear fashion due to the nonlinear behavior between clouds, ozone, surface albedo and radiation). Some discussion and acknowledgement of the potential pitfalls of this approach would be helpful in the paper. Our response. In the previous manuscript, the second proxy model was used to find out sources of the long-term variability in UV radiation at Hornsund. Now, we propose much simpler approach in the revised manuscript to solve this task. We use simulations by radiative transfer model to find combined effects of total ozone/albedo changes on surface UV. The clear-sky time series is compared to all-sky series to reveal cloud forcing on UV. Thus, parts of manuscript dealing with the performance of second proxy model have been deleted.

General comments on trend analysis: The proxy model (Eq.2) will be sensitive to clouds, as discussed in the paper. An underlying change in cloud fractions, cloud type (altitude, thermodynamic phase) over long time periods will manifest in the observed surface UV but will not be captured by the proxy model. Therefore, I find that ascribing behavior in long-term trends using the described approach somewhat dangerous, in particular given the large amount of uncertainties inherent in the approach for

empirical cloud modification. The analysis that the conclusions are drawn from should really contain uncertainty bars to guide the interpretation of the concluding statements regarding trends in ozone and cloudiness. We are aware of difficulties to estimate trends based on data having different sources and thus variable uncertainties. Simple approach (as that in the previous manuscript) of using only one ordinary least-square linear fit to all or parts of data, provides inappropriate estimate of the trend uncertainty. We propose a novel method to deal with the problem. Statistical analysis of the Monte-Carlo trend sample allows to determine the trend significance based on performance of many hypothetical time series having properties of the original time series. Please note a change of the manuscript title according to the referee #1 suggestion.

Please also note the supplement to this comment:
https://www.atmos-chem-phys-discuss.net/acp-2017-619/acp-2017-619-AC3-supplement.pdf